

# A conserved role of *bam* in maintaining metabolic homeostasis via regulating intestinal microbiota in *Drosophila*

Jiale Wang[1],[*], Yangyang Zhu[1],[*], Chao Zhang[1], Renjie Duan[1], Fanrui Kong[1], Xianrui Zheng[2] and Yongzhi Hua[1]

[1] Anhui Agricultural University, Hefei, China
[2] Zhangzhou Affiliated Hospital of Fujian Medical University, Zhangzhou, China
[*] These authors contributed equally to this work.

## ABSTRACT

**Background.** Previous studies have proven that *bag-of-marbles* (*bam*) plays a pivotal role in promoting early germ cell differentiation in *Drosophila* ovary. However, whether it functions in regulating the metabolic state of the host remains largely unknown.
**Methods.** We utilized GC-MS, qPCR, and some classical kits to examine various metabolic profiles and gut microbial composition in *bam* loss-of-function mutants and age-paired controls. We performed genetic manipulations to explore the tissue/organ-specific role of *bam* in regulating energy metabolism in *Drosophila*. The DSS-induced mouse colitis was generated to identify the role of *Gm114*, the mammalian homolog of *bam*, in modulating intestinal homeostasis.
**Results.** We show that loss of *bam* leads to an increased storage of energy in *Drosophila*. Silence of *bam* in intestines results in commensal microbial dysbiosis and metabolic dysfunction of the host. Moreover, recovery of *bam* expression in guts almost rescues the obese phenotype in *bam* loss-of-function mutants. Further examinations of mammalian *Gm114* imply a similar biological function in regulating the intestinal homeostasis and energy storage with its *Drosophila* homolog *bam*.
**Conclusion.** Our studies uncover a novel biological function of *bam*/*Gm114* in regulating the host lipid homeostasis.

# INTRODUCTION

In order to ensure the daily usage of energy and organismal fitness (*Smith et al., 2018*; *Judge & Dodd, 2020*), animals involve a wide variety of ways to maintain the metabolic homeostasis (*Judge & Dodd, 2020*; *Heindel et al., 2017*; *Hotamisligil & Erbay, 2008*). In human beings, the abnormal metabolisms of nutritional materials (namely lipid, glucose, protein, vitamin, *etc.*) have been suggested to be associated with the increased risk of many diseases (*Heindel et al., 2017*), the hallmarks of which include type 2 diabetes, cardiovascular disease, cancer, and even shortened lifespan (*Eckel, Grundy & Zimmet, 2005*). It was reported in 2016 that 39% of adults over 18 years of age worldwide were overweight, 13% were obese, and those numbers have kept growing with a relatively fast

Corresponding authors
Xianrui Zheng,
hua15551630162@163.com
Yongzhi Hua, yongzhi-
hua1112@163.com

speed during recent years, making obesity an escalating global public health problem nowadays (*Mohammed et al., 2018*). Pioneering studies have shown that the increasing incidences of energy imbalance and metabolic syndrome are resulted from nutrient excesses due to increased food consumption and reduced levels of physical activities (*Vijay-Kumar et al., 2010*; *Hall et al., 2019*). Thus, exploring functional genes that are responsible for energy storage and metabolic homeostasis can help to broaden our knowledge in obesity epidemic and contribute to the medical treatments for obesity and the related diseases (*Toubal et al., 2013*).

Since the first obese mutant (namely *adipose*) was discovered in the early 1960s (*Doane, 1960*), *Drosophila melanogaster* has been developed as a particularly useful animal model to study obesity and metabolic disease (*Suh et al., 2007*; *Teague, Clark & Doane, 1986*; *Musselman & Kuhnlein, 2018*). First, *Drosophila* share comparative anatomy of organs and cell types involved in metabolic homeostasis and develop obesity or other associated complications during caloric overload, which is highly similar to humans. Second, they share genetic similarities with humans in metabolic regulation since most genes and gene families known to function in energy storage are evolutionarily conserved. Third, they are cost-effective model organisms with relatively short lifecycles, and harbor unparalleled collection of mutants and extremely powerful genetic tools worldwide.

In *Drosophila*, it has been suggested that the fat body tissue serves as the primary lipid storage organ and the central metabolic hub (*Zheng, Yang & Xi, 2016*). It accumulates the vast majority of body fat during development by taking up lipids from the fly blood or hemolymph and esterifying them as stored triacylglyceride (TAG) and cholesterol ester when caloric overload. When energy from stored nutrient is needed for survival during metamorphosis, starvation or egg production, lipolysis will occur in the fat body (*Li, Yu & Feng, 2019*). Discoveries in *Drosophila* have demonstrated that these processes are complexly regulated by a variety of intrinsic factors in the fat bodies (*Liu et al., 2009*). However, some studies also highlighted the essential roles played by other tissues or organs such as heart, muscle, and gut in regulating lipid homeostasis. For instance, (1) the heart itself can control TAG circulation and store lipid in a non-autonomous manner via the regulation of lipoprotein metabolism (*Lee et al., 2017*). Cardiomyocyte-specific knockdown of the microsomal TAG transfer protein or its target Lpp (lipoprotein) prevents fat body TAG accumulation during high fat diet-induced obesity (*Lee et al., 2017*). (2) Muscle FoxO (class O of forkhead box transcription factors) is able to promote fat body lipogenesis remotely, inhibiting Akh expression via the secreted myokine Upd2 (Unpaired 2) (*Zhao & Karpac, 2017*). Similarly, the muscle acts as an endocrine regulator of lipid storage in the fat body by secreting activin, a member of the TGF-$\beta$ (transforming growth factor beta) cytokine family (*Yang et al., 2010*). (3) The gut microbiota and their metabolites modulate fat storage in the fly (*Wong, Vanhove & Watnick, 2016*; *Zhu et al., 2021*; *Cai et al., 2021*). Both diet and immunity play essential roles in controlling the composition of the microbiome, affecting fly health and metabolism (*Wong, Vanhove & Watnick, 2016*; *Ji et al., 2014*).

The *Drosophila bag-of-marbles* (*bam*) was first discovered in 1989, and has long been well-known as one of the key factors in determining the differentiation fate of germline

stem cells and plays essential roles in both spermatogenesis and oogenesis (*Kai, Williams & Spradling, 2005*; *Li et al., 2009*; *Shen et al., 2009*; *Yang et al., 2017*). In-depth studies further unraveled the biochemical natures of Bam protein as a ubiquitin associator (*Ji et al., 2017*) and positively regulates the deubiquitination processes of specific ubiquitinated targets. Of particular interest, *bam* has also been demonstrated to be involved in regulating the intestinal immune homeostasis and the host lifespan (*Ji et al., 2019*). As one type of the most common post-translational modifications of proteins, ubiquitination has been indicated to be involved in nearly every aspect of biological processes within animal cells (*Akutsu, Dikic & Bremm, 2016*). Thus, it is natural for one to hypothesize much broader functions of *bam* in terms of its critical roles in regulating ubiquitin-dependent protein modifications and related biological activities.

In this study, we demonstrate that loss of *bam* leads to an increased storage of lipids throughout the whole adult life of *Drosophila*. We observe detectable *bam* expressions at both mRNA and protein levels in various tissues including guts and fat bodies, besides the reproductive system where it is significantly abundant. Of interest, by utilizing UAS/Gal4 system, specific silence of *bam* in intestinal cells but neither the fat bodies nor the reproductive organs results in metabolic dysfunction, suggesting that *bam* contributes to metabolic regulation of the host through its essential role in intestines. This notion is further supported by the results obtained from rescue experiments. Further, we show that *bam* executes its biological functions in metabolic regulation probably via controlling the gut microbiota. Moreover, knocking out of *Gm114*, the mammalian homolog of *Drosophila bam*, causes intestinal dysfunction, microbial dysbiosis, and obese phenotype in mice. Taken together, our studies suggest an evolutionarily-conserved role of *bam* in regulating lipid storage and metabolic homeostasis.

## MATERIALS & METHODS

### *Drosophila* strains

Fly stocks were reared with standard culture mediums (6.65% cornmeal, 7.15% dextrose, 5% yeast, 0.66% agar supplemented with 2.2% nipagin and 3.4 ml/l propionic acid). The $w^{1118}$ strain was used as the control and the tool strain for isogenization. The P{*NP1-Gal4* } (gut driver), P{*Nos-Gal4* } (germline driver), P{*Ppl-Gal4* } (fat body driver), P{*Elav-Gal4* } (neuron driver), $bam^{\Delta 86}$ (*bam* loss-of-function mutant), P{*Uasp-artmiR-bam* } (*bam KD*), and P{*Uasp-bam*} strains were obtained from the Bloomington *Drosophila* Stock Center and described previously (*Guo et al., 2014*; *Ji et al., 2017*).

### Mice

*Wild-type* (*WT, C57BL/6* ) and *Gm114* knock out (*Gm114 KO*) mice (*Tang, Ross & Capel, 2008*) were purchased from Nanjing Myris Biotechnology Co. and maintained under pathogen-free conditions in the animal care center of Anhui Agricultural University (22 ± 2 °C, 12/12 h light/dark cycle). Each mouse has been taken as the experimental unit, and the mice used in the experiments were randomly selected. All mice were carried out in accordance with the "Regulations on Animal Management" of the Ministry of Health of the People's Republic of China and the plan approved by the Laboratory Animal Center

of Anhui Agricultural University (SYXK 2020-007). Intraperitoneal injection of sodium pentobarbital was used to euthanize the mice at the end of the experiments. The operating procedures involving animals in animal experiments have been approved by experimental Animal Center of Anhui Agricultural University, and the IACUC approval number is AHAU 210429.

## Antibodies
Western blotting assays were performed according to methods described previously (*Hua et al., 2022*). The following antibodies were utilized for Western blotting: Bam (mouse, 1:3000, DSHB), Actin (mouse, 1:5000, Cat#A4700; Sigma), and Goat anti-Mouse IgG H&L (1:3000, Cat#ab150078; Abcam).

## Examination of total lipids via GC–MS
The GC-MS assays were performed as previously described (*Papadimitropoulos et al., 2018*; *Tennessen et al., 2014*). Briefly, indicated flies were added with 800 µl prechilled 90% methanol containing 1.25 µg/ml succinic-d4 acid (293075; Sigma) and 6.25 µg/ml U-13C, U-15N amino acid mix (CDNLM-6784; Cambridge Isotope). Samples were then homogenized for 30 s at 6.45 m/s and incubated at −20 °C for 1 h to enhance protein precipitation, followed by centrifugation at 20,000 g for 5 min at 4 °C to remove the resulting precipitate. The supernatant was transferred to a 1.5 ml microfuge tube and the solvent removed with a Speed-Vac (Genevac). The GC-MS detection and analysis software was opened to create the experimental method (6890 Agilent GC with Leco pegasus IV time-of-flight MS instrument). 0.5ul of mixed reagent was drawn with a micro syringe and the sample was injected through the sample inlet of the gas chromatograph. Last, we use the BinBase database software to perform analysis.

## Measurements of metabolites
The analyses of metabolites including TAG (triacylglyceride), Gly (glycogen), and Glu (glucose) were performed to examine the metabolic profiles of indicated flies (*Tennessen et al., 2014*; *Fan et al., 2017*). Briefly, flies were collected and frozen at −80 °C until use. 5 flies per group were homogenized in $1 \times$ PBST and centrifuged at 1,000 g for 2 min. Proper aliquot of the supernatant was used to measure protein concentration by the Bradford assay (BCA1; Sigma), or TAG level by working reagent (T2449; Sigma), or Gly and Glu levels using HK kit (MAK091; Sigma). The total amounts of TAG, Gly, and Glu were normalized to the total protein level.

## Cafe and Fluorescein feeding assays
The cafe assay (*Cai et al., 2021*) was performed to monitor the levels of food consumption of indicated flies. Briefly, two labeled calibrated glass micropipettes (5 µl, VWR) were filled with liquid medium and were inserted through the cap via truncated 200 µl pipette tips. 10 flies per vial were subjected to the long-term experiments (six independent replicates for each group) under a 12 h light/12 h dark cycle in a room kept at 25 °C and 60% humidity. Each experiment included an identical Cafe chamber without flies to determine evaporative losses (typically <10% of ingested volumes), which were subtracted from experimental readings.

Flies were put into vials with meals that contained 50 μM Fluorescein (F6377; Sigma) for the Fluorescein feeding assay. 10 flies were collected and lysed with 10 mM K-phosphate buffer (pH 6.0) 2 h later and 6 independent replicates for each group. Lysates were put to a 96-well plate (Corning) after centrifugation, and the plate reader (Tecan) was used to conduct the analysis. Excitation at 480 nm and emission at 521 nm were used to measure the fluorescein levels. Equal volume of K-phosphate buffer was utilized as the baseline control in this assay.

### RT-qPCR assays

Total RNA from dissected gut, head, fat body, reproductive organ, thorax, and the whole body were extracted using Trizol reagent (15596018; Thermo Fisher) according to the manufacturer's instructions. cDNA was synthesized using a first-strand cDNA synthesis kit (Transgene, AT301-02). Quantitative PCR was performed in triplicate using SYBR Green Master Mix (A25780; Thermo Fisher) on a Light Cycler 480 following previously published methods (*Cai et al., 2018*). Relative gene expression levels were normalized to the levels of *rp49* in each sample. Primers used for RT-qPCR assays are as follows.

Bam-Fw: CGAGGAAAGCCACTTGTGAG;
Bam-Rv: GTTGCAAGCAATCCAAACCG;
Rp49-Fw: CACGATAGCATACAGGCCCAAGATCGG;
Rp49-Rv: GCCATTTGTGCGACAGCTTAG;
TNF $\alpha$-Fw: TCCCCAAAGGGATGAGAAGTT;
TNF $\alpha$-Rv: GTTTGCTACGACGTGGGCTAC;
IL6-Fw: TCGGAGGCTTAATTACACATGTTCT;
IL6-Rv: TGCCATTGCACAACTCTTTTCT.

### Measurements of gut microbiota

Intestines (from indicated flies or mice) were collected and subjected to genomic DNA extraction according to the manufacturer's protocols (E.Z.N.A.® Insect DNA Kit). Quantitative PCR assays were further performed to determine the population of the total intestinal microbiota, Bacilli, Gammaproteobacteria, or Alphaproteobacteria using specific primers. The expression levels of *actin* or *GAPDH* were used as internal controls for fly or mice samples, respectively. The sequences for the primers are outlined below.

16S-Fw: AGAGTTTGATCCTGGCTCAG;
16S-Rv: CTGCTGCCTYCCGTA;
Ba-Fw: CGACCTGAGAGGGTAATCGGC;
Ba-Rv: GTAGTTAGCCGTGGCTTTCTGG;
Ga-Fw: GGTAGCTAATACCGCATAACG;
Ga-Rv: TCTCAGTTCCAGTGTGGCTGG;
Al-Fw: CCAGGGCTTGAATGTAGAGGC;
Al-Rv: CCTTGCGGTTCGCTCACCGGC;
Actin-Fw: TTGTCTGGGCAAGAGGATCAG;
Actin-Rv: ACCACTCGCACTTGCACTTTC;
GAPDH-Fw: ATGACATCAAGAAGGTGGTG;
GAPDH-Rv: CATACCAGGAAATGAGCTTG.

### Rearing *Drosophila* under axenic conditions

To raise flies under axenic conditions, we prepared *Drosophila* foods supplemented with antibiotics (500 µg/ml ampicillin, 50 µg/ml tetracycline, and 200 µg/ml rifamycin) as previously described (*Koyle et al., 2016*). The vials containing foods were autoclaved for 30 min, followed by 12 h irradiation by a radioactive cesium source.

### DSS-induced colitis in *Gm114* mice and HE staining

To induce colitis in mice, 2% (w/v) DSS (molecular weight 36,000-50,000; MP Biomedicals, 9011-18-1) was added to the drinking water for 7 d. For the measurement of weight, mice were further treated with normal drinking water for 14 d and measured for daily weight. For HE staining of colon, mice were sacrificed on d 8, and the entire colon was excised. The clean colons opened longitudinally were fixed with 10% formalin. Small pieces of fixed colon tissues were dehydrated and embedded in paraffin (Paraplast Tissue Embedding Medium, LEICA) using a modular tissue embedding system (Leica EG1150 H). The 5 µm sections were cut using a fully automated rotary microtome (Leica RM2255) and mounted on positively charged slides (Adhesion Microscope Slides; CitoGlas; Haimen, Jingsu, China). HE staining was carried out using a kit (Boster Biological Technology Company, Pleasanton, CA, USA).

### Statistical analysis

For all statistical analyses, data are shown as means and standard errors. Statistical significances were determined using the two-tailed Student's *t* test (in Figs. 1A–1D, 2C–2E, 2G–2H, 3A–3H, 4A, 4C–4F, 5C, Figs. S1A–S1B, Figs. S2A–S2B) or LogRank test in the PASW Statistics 18 software in Fig. 5B. The *p* value of less than 0.05 was considered statistically significant. * $p < 0.05$; ** $p < 0.01$; *** $p < 0.001$; ns, not significant. Results from Figs. 1A–1D, 2A, 2C–2E, 2G–2H, 3A–3H, Figs. S1A–S1B, Figs. S2A–S2B were obtained from 3 biological replicates. Data from Figs. 4C, Figs. 4E and 4F were obtained from 7 biological replicates. In Figs. 4A and 4D, the numbers of experimental mice were 38 for *WT,fig35* for *Gm114 KO*. In Figs. 5B and 5D, the numbers for DSS-treated *WT* and *Gm114 KO* mice were 15 and 17, respectively.

## RESULTS

### Loss of *bam* leads to excessive energy storage in *Drosophila*

To explore the potential involvements of *bag-of-marbles* (*bam*) in impacting the metabolic states in *Drosophila*, we utilized the $bam^{\Delta86}$ homozygous mutant ($bam^{-/-}$), heterozygous mutant ($bam^{+/-}$), and $w^{1118}$ ($bam^{+/+}$, referred as *wild-type* control), and performed gas chromatography and mass spectrometer (GC-MS) analyses to assess the levels of total lipids. As shown in Fig. 1A, *bam* loss-of-function mutant flies (at the age of 10-day) exhibited elevated (by ∼20%) levels of total lipids relative to controls. The occurrences of total lipids in *bam* mutants increased more significantly (by ∼40% and ∼70%) from samples at the ages of 30-day and 50-day, respectively, when compared to those of the indicated controls (Fig. 1A), implying that silencing of *bam* likely lead to energetic increases in *Drosophila*.

To further test our hypothesis, we first examined the index of triacylglyceride (TAG), which is the main form of lipid storage and has long been used to define obesity in fruit

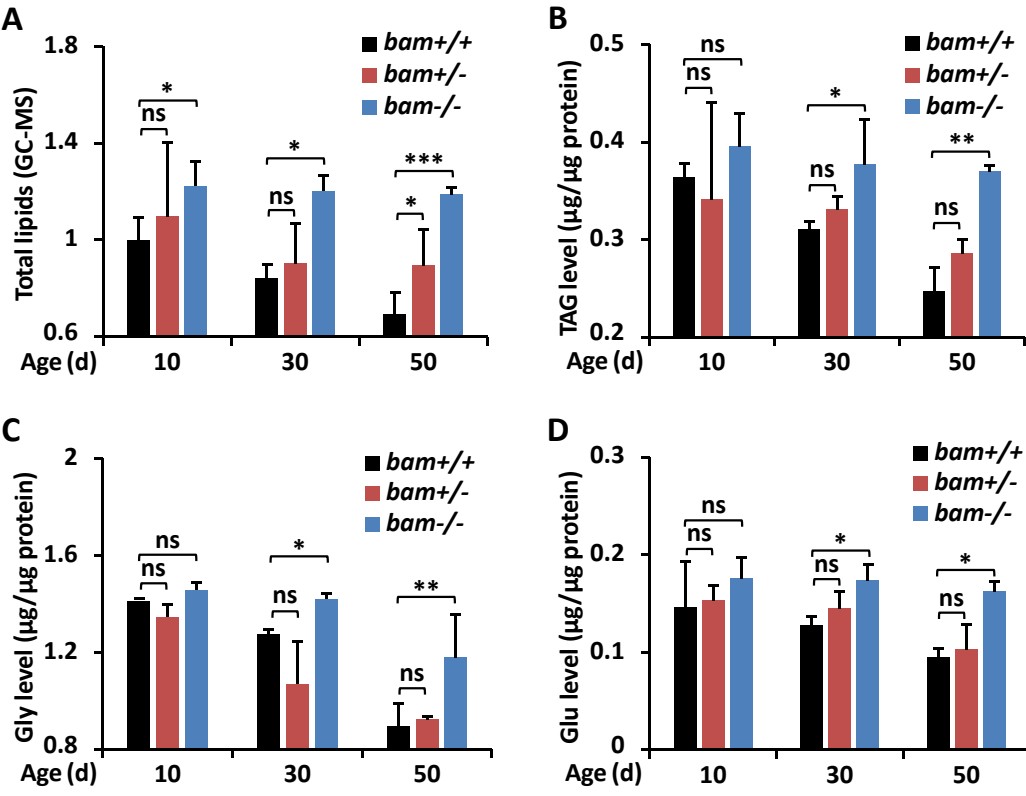

**Figure 1** **Loss-of-function of *bam* leads to excessive energy storage in *Drosophila*.** (A–D) $w^{1118}$ ($bam^{+/+}$), $bam^{\Delta 86}$ heterozygous ($bam^{+/-}$) and homozygous ($bam^{-/-}$) mutants were raised under normal conditions. At indicated ages (10-day, 30-day, and 50-day, respectively), flies were subjected to metabolic assays. The levels of total lipids (A), TAG (B), Gly (C), and Glu (D) were analyzed and shown. Error bars represent SD ($n = 3$). The two-tailed Student's t test was used to analyze statistical significance. ns, not significant, * $p < 0.05$, ** $p < 0.01$, *** $p < 0.001$.

fly, as in humans (*Arrese & Soulages, 2010*; *Bharucha, Tarr & Zipursky, 2008*; *Carley & Lewandowski, 2016*). Even though we observed overall decreases of TAG in all samples with indicated genotypes during aging, which was consistent with the previous observations (*Johnson & Stolzing, 2019*), the levels of TAG in *bam* mutant adults were significantly higher than those in the controls (Fig. 1B). We then sought to quantify the levels of glycogen (Gly), which represents the state of energy storage in the host (*Yamada et al., 2018*). Consistently, we observed that mutation of *bam* resulted in marked augmentations (by ~10% to ~30%) of Gly abundances throughout the whole adult life (Fig. 1C). We last quantified the levels of the basic metabolite glucose (Glu), since previous studies have shown that Glu is one of the primary forms of circulating carbohydrates in *Drosophila*, and serves as an essential energy source and substrate for biosynthetic reactions of glycogen (*Fan et al., 2017*). As illustrated in Fig. 1D, the circulating Glu was approximately equivalent from samples of controls and flies with only one copy of *bam* mutation at various stages of age. However, this sugar accumulated more in age-paired *bam* homozygous adults (Fig. 1D). Taken together, our

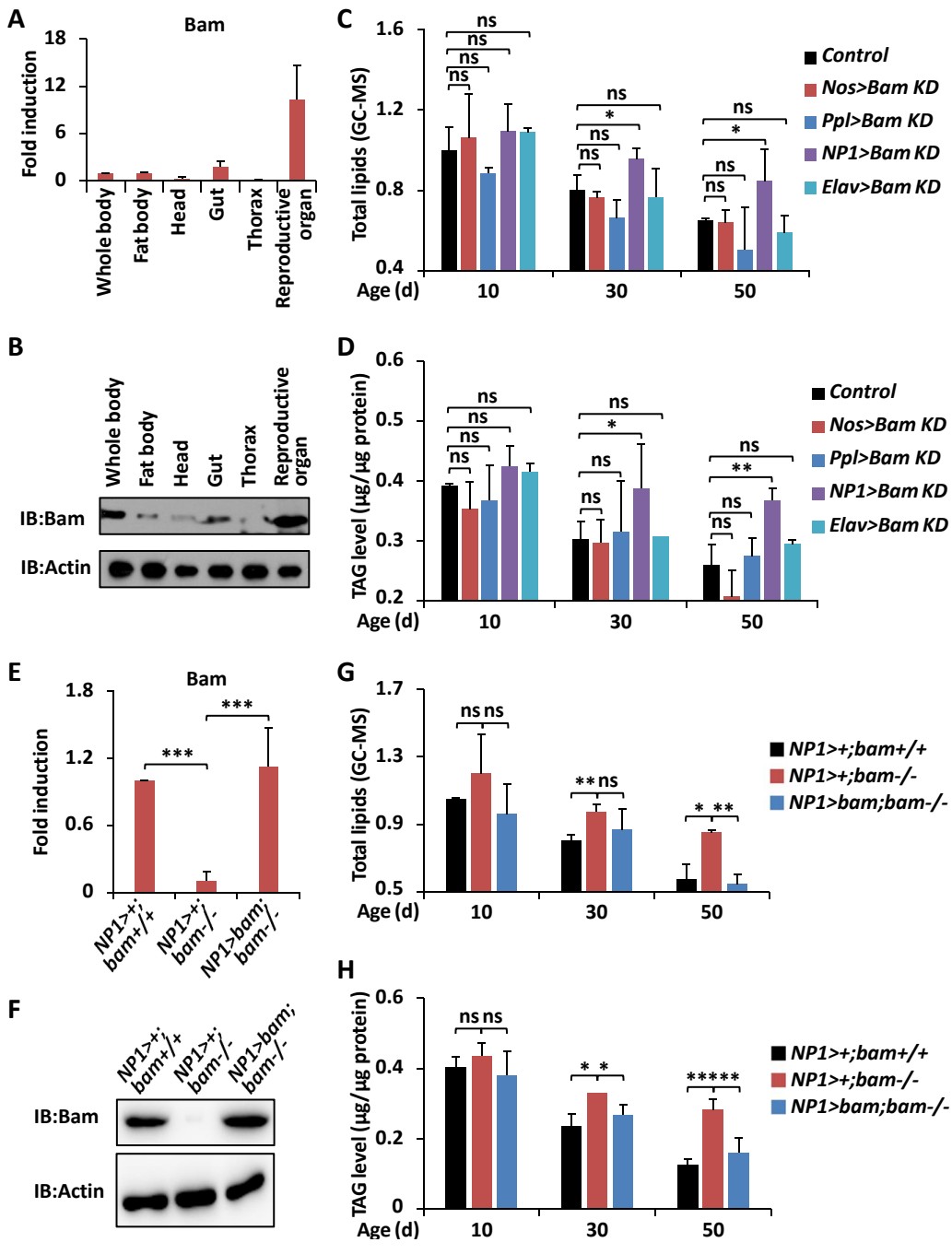

**Figure 2** ***bam* functions in guts to impact *Drosophila* metabolism.** (A and B) The whole flies or specific parts including fat body, head, gut, thorax, and reproductive organ were dissected and lysed for RT-qPCR assays to measure the expression profiles of *bam* mRNA (A) or for Western blot assays to measure the expression levels of Bam protein (B). In A, error bars represent SD ($n = 3$).(C and D) Flies including *Nos >Bam KD*, *Ppl >Bam KD*, *NP1 >Bam KD*, *Elav >Bam KD*, and the control (*Bam KD/ +*) were harvested at indicated ages (10-day, 30-day, and 50-day, respectively) and subjected to GC-MS analysis (C) or metabolic assays (D) to monitor the levels of total lipids (C) or TAG (D). Error bars represent SD ($n = 3$). (continued on next page...)

**Figure 2 (...continued)**
The two-tailed Student's t test was used to analyze statistical significance. ns, not significant, * $p < 0.05$, ** $p < 0.01$.(E and F) Guts were dissected from $NP1 > +; bam^{+/+}$, $NP1 > +; bam^{-/-}$, and $NP1 > bam; bam^{-/-}$, followed by RT-qPCR assays (E) or Western blot assays (F) to measure the expression profiles of *bam* at mRNA (E) or protein (F) levels, respectively. Error bars represent SD ( $n = 3$). The two-tailed Student's t test was used to analyze statistical significance. *** $p < 0.001$. (G and H) $NP1 > +; bam^{+/+}$, $NP1 > +; bam^{-/-}$ and $NP1 > bam; bam^{-/-}$ flies were harvested at various ages (10-day, 30-day, and 50-day, respectively) and subjected to GC-MS analysis (G) or metabolic assays (H) to detect the levels of total lipids (G) or TAG (H). Error bars represent SD ($n = 3$). The two-tailed Student's t test was used to analyze statistical significance. ns, not significant, * $p < 0.05$, ** $p < 0.01$, *** $p < 0.001$.

results indicate that *bam* largely plays a critical role in regulating the metabolic homeostasis in *Drosophila*.

## Silencing of *bam* is dispensable for impacting food consumption in *Drosophila*

We sought to determine whether *bam* is involved in impacting food consumption of *Drosophila*, since previous evidence has shown that increased nutrient absorption and storage might be a consequence of elevated intake of foods (*Kohyama-Koganeya, Kurosawa & Hirabayashi, 2015*; *Branch & Shen, 2017*). To do this, we first performed the cafe assay, a widely used method for directly and accurately measuring feeding rates by using capillary feeds containing liquid media (*Ja et al., 2007*). However, we failed to observe any obvious increases in food intake of *bam* mutant adults, when compared with the age-matched controls (Fig. S1A). To further confirm these findings, we subsequently used the Fluorescein as a food tracer (*Wang et al., 2005*), and conducted natural feeding experiment as previously described (*Danilov et al., 2015*). We obtained consistent results, as there were no apparent variances among the levels of consumed-foods from the age-paired *bam* homozygous and heterozygous mutant flies, and the $w^{1118}$ controls (Fig. S1B). Collectively, our results suggest that the increased energy storage caused by silencing of *bam* is not likely through abnormal food consumption in *Drosophila*.

## Prevention of *bam* in intestines results in host obesity

In order to investigate how *bam* contributes to regulating the metabolic states in *Drosophila*, we first sought to explore whether this regulatory relationship is organ- or tissue-dependent. Numerous pioneering studies have shown that *bam* is highly expressed in reproductive organs (namely ovary and testis), and both necessary and sufficient for the differentiation of germline stem cells and cystoblasts (*Ohlstein & McKearin, 1997*; *McKearin & Spradling, 1990*; *McKearin & Ohlstein, 1995*). Significantly, *bam* was reported to exist in intestinal cells, contributing to the maintenance of gut homeostasis and host lifespan (*Ji et al., 2019*). Therefore, we determined to examine the expression profiles of *bam* in various tissues. We dissected $w^{1118}$ adult flies and separately collected tissues including fat body, head, gut, thorax, and reproductive organ. The reverse transcriptional quantitative polymerase chain reaction (RT-qPCR) and Western blotting assays further displayed that *bam* was highly abundant in reproductive organs, moderately expressed in intestinal and fat body cells, and relatively low expressed in tissues of head and thorax (Figs. 2A and 2B).

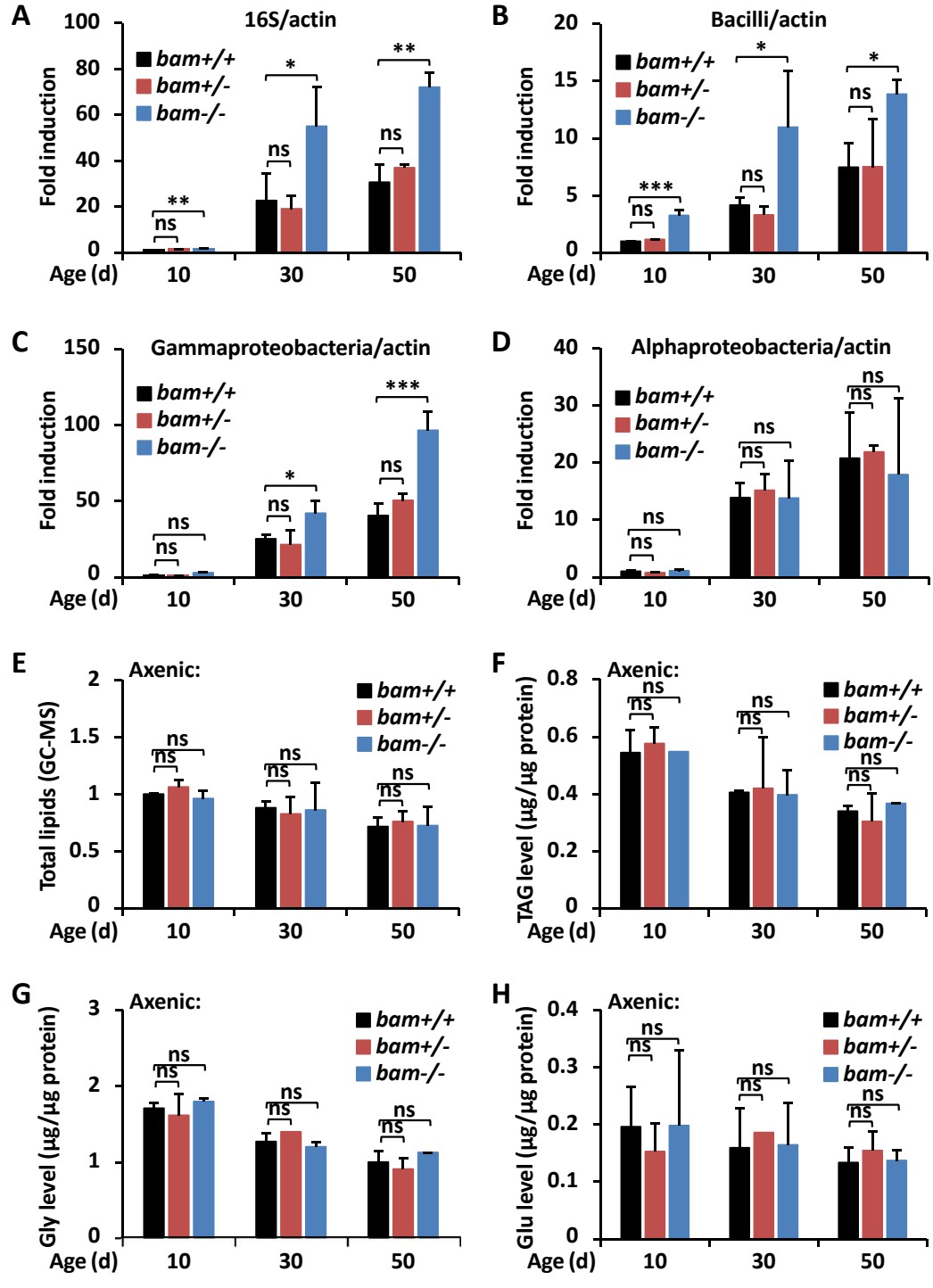

**Figure 3** ***bam*** **regulates** ***Drosophila*** **metabolic homeostasis through targeting gut microbiota.** (A–D) Guts were dissected from $w^{1118}$ ($bam^{+/+}$), $bam^{\Delta 86}$ heterozygous ($bam^{+/-}$) and homozygous ($bam^{-/-}$) mutant flies at indicated ages, followed by DNA extractions and qPCR assays to monitor the populations of total microbiota (A), Bacilli (B), Gammaproteobacteria (C), or Alphaproteobacteria (D). Error bars represent SD ($n = 3$). 

## Figure 3 (…continued)

The two-tailed Student's t test was used to analyze statistical significance. ns, not significant, * $p < 0.05$, ** $p < 0.01$, *** $p < 0.001$.(E–H) $w^{1118}$ ($bam^{+/+}$), $bam^{\Delta 86}$ heterozygous ($bam^{+/-}$) and homozygous ($bam^{-/-}$) mutant flies were reared under axenic conditions. At various ages, flies were subjected to metabolic analysis to quantify the levels of total lipids (E), TAG (F), Gly (G), or Glu (H). Error bars represent SD ($n = 3$). The two-tailed Student's t test was used to analyze statistical significance. ns, not significant.

We then utilized the widely-used UAS/Gal4 driver system (*Brand & Perrimon, 1993*) to down-regulate *bam* expression in specific organs. As shown in Fig. 2C, the P{*Nos-Gal4* } (germline driver), P{*Ppl-Gal4* } (fat body driver), P{*NP1-Gal4* } (gut driver), and P{*Elav-Gal4* } (neuron driver) were crossed with *Bam KD* (*Ji et al., 2017*) to generate *Nos>Bam KD*, *Ppl>Bam KD*, *NP1>Bam KD*, and *Elav>Bam KD*, respectively. When we investigated the levels of total lipids and TAG from those flies throughout the whole adult life, we found that down-regulation of *bam* in either the reproductive organ, or the fat body, or the nervous system was dispensable for affecting the lipid metabolism in *Drosophila* (Fig. 2C). In particular, prevention of *bam* expression in intestinal cells resulted in markedly elevated (by ∼20% to ∼40%) storages of energy relative to age-paired (30-day and 50-day, respectively) controls (Figs. 2C and 2D), even though there were no significant statistical variances from samples collected at the early adult stage (10-day). These results were consistent with our previous observations in *bam* mutant flies (Fig. 1), indicating that *bam* functions in the intestinal cells to control the metabolic homeostasis of the host during aging.

### Restoring *bam* in guts rescues energetic increase in *bam*$^{\Delta 86}$ homozygotes

To further verify the regulatory role and ability of intestinal *bam* for the lipid storage and metabolism, we next performed rescue experiments. As shown in Figs. 2E and 2F, the expression levels of *bam* were almost comparable in gut samples from both the control ($NP1 > +$; $bam^{+/+}$) and the rescue groups ($NP1>bam$; $bam^{-/-}$). When we collected these flies and performed total lipid and TAG analyses, we found that over-expression of *bam* in gut cells nearly fully rescued the excessive energy storages in $bam^{\Delta 86}$ homozygotes (Figs. 2G and 2H). Collectively, our results suggest that *bam* is responsible for controlling lipid metabolism through its essential role in the intestinal tract in *Drosophila*.

### *bam* contributes to metabolic homeostasis via regulating gut microbiota

Numerous pioneering studies have shown that dysregulation of gut microbiota leads to metabolic disorders both in invertebrates and vertebrates (*Vijay-Kumar et al., 2010*; *Wong, Vanhove & Watnick, 2016*; *Ji et al., 2019*; *Guo et al., 2014*; *Lee & Hase, 2014*; *Clark et al., 2015*), suggesting a pivotal role of the commensal intestinal bacteria in affecting the host metabolism. To further address whether *bam* is involved in controlling gut microbiota of *Drosophila*, we first examined the total bacterial load in fly guts at various ages. As shown in Fig. 3A, $bam^{\Delta 86}$ homozygotes displayed higher (more than two-fold) populations of intestinal bacteria compared to those of $bam^{\Delta 86}$ heterozygotes or $w^{1118}$ controls. We next performed qPCR assays using different primers determining the occurrences of bacteria

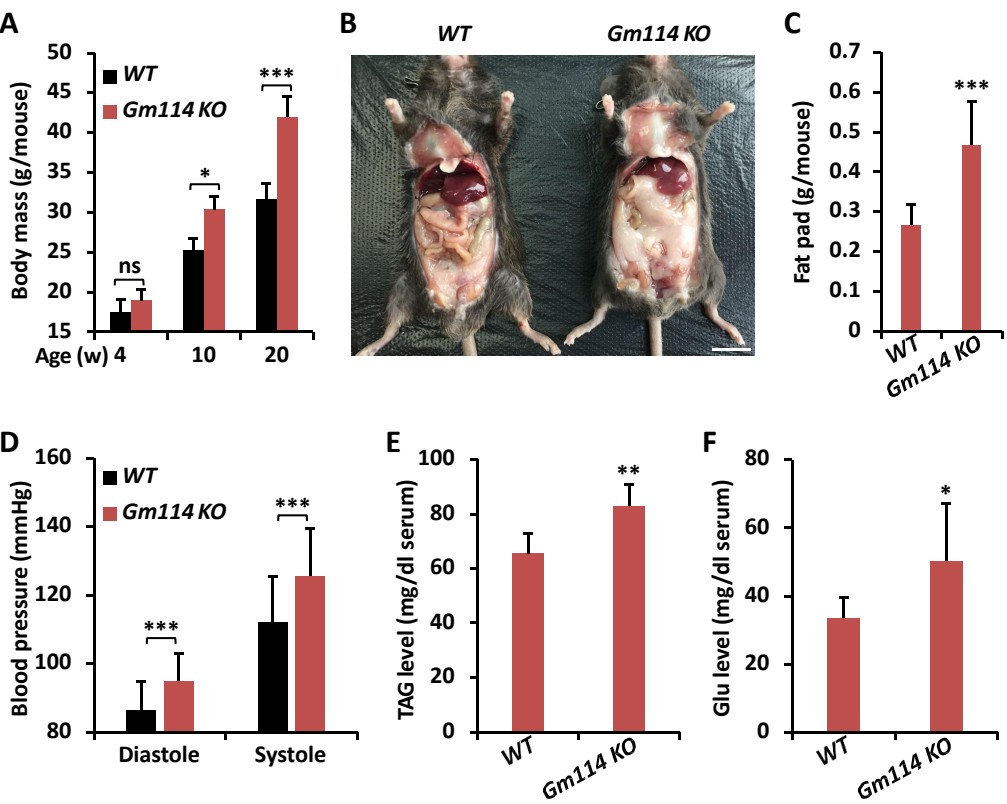

**Figure 4** *Gm114 KO* **mice display obesity.** (A) *Gm114 KO* ($n = 35$) mice gained more weights than age-paired *WT* ($n = 38$) controls. Error bars represent SD. The two-tailed Student's t test was used to analyze statistical significance. ns, not significant, * $p < 0.05$, *** $p < 0.001$. (B) Abdominal photograph of representative male mice at the age of 20-week. Scale bars, one cm. (C–F) 20-week *Gm114 KO* and *WT* mice were subjected to quantification of fat pad (C), examination of blood pressure (D), calculation of TAG (E), or Glu (F). Error bars represent SD. The two-tailed Student's t test was used to analyze statistical significance. * $p < 0.05$, ** $p < 0.01$, *** $p < 0.001$.

including Bacilli, Gammaproteobacteria, and Alphaproteobacteria, which have been shown to be the main causes of intestinal dysfunction in *Drosophila* (*Clark et al., 2015*). Indeed, we observed marked elevations (by ~60% to ~220%) of the populations of both Bacilli and Gammaproteobacteria but not the Alphaproteobacteria in *bam*$^{\Delta 86}$ homozygous mutants, when compared to those in the controls (Figs. 3B–3D). These results imply that loss of *bam* contributes to the dysbiosis of microbiota in *Drosophila* guts, thus may leading to increased lipid storage. Then, we raised *bam* mutants and the control flies under axenic condition (*Koyle et al., 2016*) to eliminate the influence of gut microbiota. As shown in Figs. 3E–3H, *bam* mutants and the control flies displayed similar metabolic profiles in the context of total lipids and various metabolites, when the gut microbiota was abolished. Collectively, our results suggest that loss of *bam* leads to metabolic disorder, which may be related to the imbalance of intestinal microbiota in *Drosophila*.

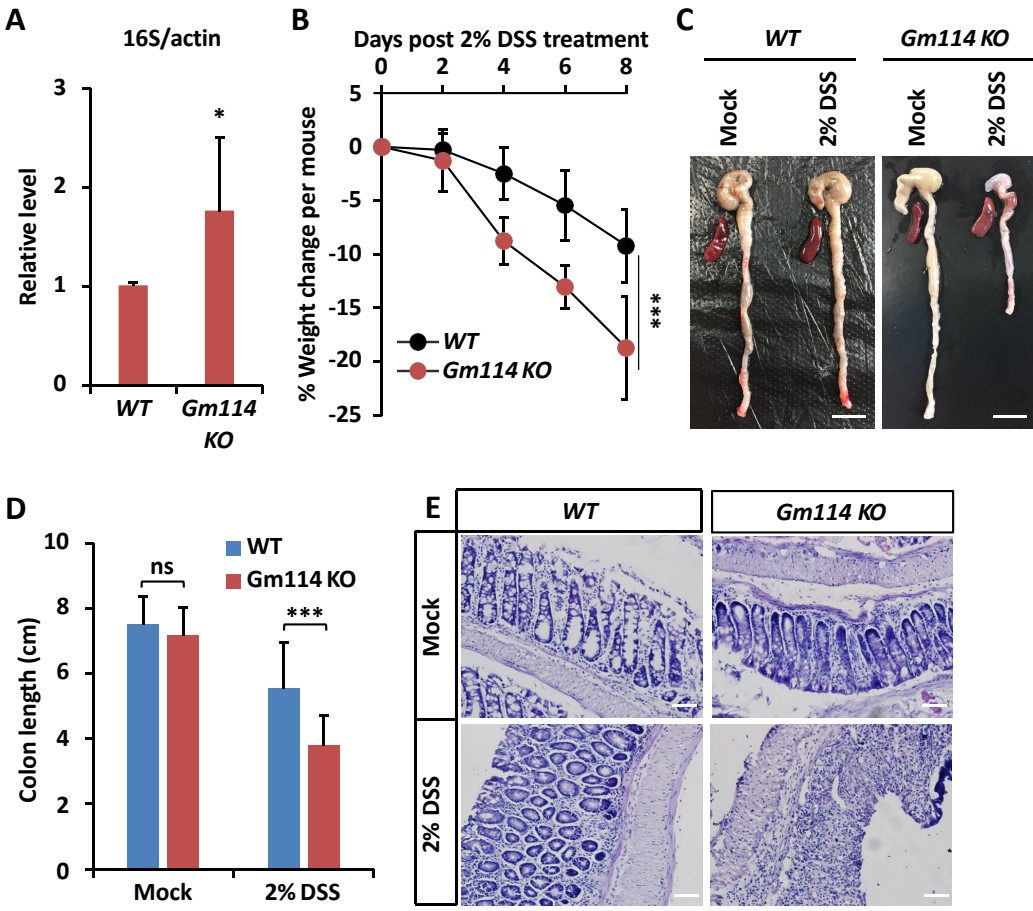

**Figure 5** *Gm114 KO* **mice are susceptible to DSS-induced colitis.** (A) qPCR assays were performed to monitor the gut microbial populations of *Gm114 KO* and *WT* mice. Error bars represent SD ($n = 3$). The two-tailed Student's t test was used to analyze statistical significance. * $p < 0.05$.(B) Weight changes of *Gm114 KO* ($n = 17$) and *WT* ($n = 15$) mice following DSS-treatment. Error bars represent SD. The LogRank test in the PASW Statistics 18 software was used to analyze statistical significance. *** $p < 0.001$. (C and D) Representative photos (C) showing the morphological changes of colon and spleen of indicated mice. Scale bars, one cm. Colon lengths were calculated and shown in (D). Error bars represent SD. The two-tailed Student's t test was used to analyze statistical significance. ns, not significant, *** $p < 0.001$. (E) Indicated colons were collected and subjected to HE staining. Representative histological images were shown. Scale bars, 40 μm.

## *Gm114* regulates lipid metabolism in mice

Previous study has shown that *Gm114* displays similar expression profile in mouse testis to that of its *Drosophila* homologue *bam* (*Tang, Ross & Capel, 2008*). However, silencing of *Gm114* hardly affected the development of germ cells and the fertility of mice to a large extent, which indicates that there are no functional parallels between *Gm114* and *bam* in regulating spermatogonial differentiation (*Tang, Ross & Capel, 2008*). When we raised *Gm114* knockout (*Gm114 KO*) mice under normal condition, we observed that they gained more weights than the *wild-type* (*WT*) controls and this phenomenon was more significant in 10- and 20-week animals (Fig. 4A). We also found that *Gm114 KO* mice

displayed obvious fat accumulation (mostly the white adipose tissue) in the abdominal cavity covering most of the internal organs (Figs. 4B and 4C).

We further performed a series of examinations detecting the physiological profiles of these animals. As shown in Fig. 4D, higher blood pressures were observed in *Gm114 KO* mice, when compared to those of the age-paired *WT* controls. Moreover, we noted marked increases in the host serum levels of TAG and Glu when *Gm114* was knocked out (Figs. 4E and 4F), suggesting mammalian *Gm114* plays a similar regulatory role on lipid metabolism to that of its *Drosophila* homologue *bam*.

## Loss of *Gm114* leads to intestinal dysfunction in mice

We then sought to explore whether *Gm114* is also involved in modulating intestinal homeostasis. To do this, we first performed qPCR assays to determine whether the total population of gut microbiota was affected by silencing of *Gm114*. As shown in Fig. 5A, the level of commensal bacterium in *Gm114 KO* guts was markedly increased (by ∼70%) when compared to that of the *WT*. We next observed that the mRNA levels of inflammation markers such as *TNF α* and *IL6* were elevated by knocking out of *Gm114* (Figs. S2A and S2B). Then, we administered *Gm114 KO* mice and *WT* controls with 2% DSS in drinking water for 7 days to induce a colitis model. We indeed observed an apparent loss of body weight in the *WT* mice and a faster leveling off in the weight of *Gm114 KO* mice (Fig. 5B). Of note, when we dissected the intestines from *Gm114 KO* and *WT* mice, we found that the DSS treatment-induced reduction of colon length was more evident in *Gm114 KO* mice than in WT controls, though the colon lengths were comparable among mock-treated animals (Figs. 5C and 5D). To further analyze the histology of the colons, we carried out hematoxylin-eosin (HE) staining assays. Our results showed that the intestinal injury of *Gm114 KO* mice was more severe than that of the *WT* mice: the intestinal cell morphology was completely lost and tissue was atrophied after being treated with 2% DSS (Fig. 5E). Most of our observations on DSS-treated *Gm114 KO* mice are consistent with what were found in a recent study (*Sun et al., 2022*). Taken together, these results strongly suggest that *Gm114* is essential for maintaining the intestinal homeostasis in mice.

## DISCUSSION

In this study, we report a novel biological function of *bam* in modulating lipid storage and energy metabolism in *Drosophila*. We observed that mutation of *bam* leads to increased lipid storage and dysbiosis of intestinal microbiota; deletion of the gut commensal bacteria nearly fully rescues the metabolic abnormalities by loss of *bam*. Further investigations in mice showed that knocking out of *Gm114* results in obese phenotype and dysregulation of gut microbiota, suggesting a similar role with its *Drosophila* homologue *bam*. Collectively, our findings suggest that *bam/Gm114* regulates metabolic homeostasis in an evolutionarily-conserved manner, and this regulatory mechanism may be related to the gut microbiota.

A series of pioneering studies have proven that *bam* plays a vital ON and OFF switch role in determining the differentiation fate of germline stem cells in *Drosophila* ovaries (*Li et al., 2009*; *Shen et al., 2009*; *Ohlstein & McKearin, 1997*; *McKearin & Spradling, 1990*; *McKearin & Ohlstein, 1995*). In 2017, a study exploring the biochemical nature of Bam protein,

convincingly showed that Bam is a ubiquitin-binding protein and exerts its essential biological functions via ubiquitination-involved regulatory manner with the help of a co-factor (*Ji et al., 2017*). In fact, previous studies suggested that as the mammalian homologue of *Drosophila bam*, *Gm114* displays similar expression patterns in the reproductive systems, but is dispensable for regulating germline development (*Tang, Ross & Capel, 2008*). Nevertheless, we observed that *Gm114 KO* mice displayed some obese phenotypes and intestinal dysbiosis, which was also shown in a recent study (*Sun et al., 2022*). In this regard, we would like to propose that *bam/Gm114* is essential for regulating metabolic homeostasis of the host, the manner of which is conserved in the long history of animal evolution.

How *bam* contributes to metabolic regulation? It is well-known that abnormal food intake can cause metabolic defects in both invertebrates and vertebrates (*Smith et al., 2018*; *Hotamisligil & Erbay, 2008*; *Eckel, Grundy & Zimmet, 2005*; *Hall et al., 2019*; *Musselman & Kuhnlein, 2018*; *Zheng, Yang & Xi, 2016*; *Liu et al., 2009*). Indeed, we exclude the effect of *bam* on *Drosophila* food consumption by conducting the cafe assay and the natural feeding experiment. Based upon our tissue-specific observations that *bam* controls the host metabolic state through its essential role in the gut, where a large population of commensal bacteria exist (*Clark et al., 2015*), we highly hypothesize that *bam* impacts the intestinal microbiota to affect the host metabolic homeostasis. Our investigations both in *Drosophila* and mice indeed suggest a tight connection between *bam/Gm114* and the maintenance of bacterial robustness in the gut. Moreover, deletion of intestinal microbiota markedly reverses the metabolic overload by loss of *bam*, which suggests that *bam* may play an important role in controlling host metabolism in a gut microbiota-dependent manner. Importantly, we observed that silencing of *bam* preferentially impacts the population of Bacilli and Gammaproteobacteria other than Alphaproteobacteria, implying *bam* likely specifically contributes to the regulation of some commensal bacterium. Indeed, we noted that Gammaproteobacteria expansion is a hallmark of intestinal barrier dysfunction in *Drosophila*, suggesting our observation regarding increased Gammaproteobacterial population in *bam* mutants is consistent with dysregulation of gut epithelial barrier in recent findings (*Ji et al., 2019*). Based upon previous literature and our current findings, we hypothesize that *bam*-mediated gut microbiota is involved in the controlling of the intestinal homeostasis and the host metabolism. Nevertheless, it still remains unclear how *bam* leads to dysbiosis of Bacilli and Gammaproteobacteria to further impact the metabolic state in *Drosophila*. It is of significant interest to investigate the biological functions of these commensal bacteria (or their metabolites) and the underlying mechanisms in regulating *Drosophila* metabolic homeostasis. Meanwhile, it would be very worthwhile to generate intestine-specific *Gm114* knock out mouse to explore the potential regulatory relationship between *Gm114* and the commensal gut microbiota and the underlying mechanism in the future.

Regarding the underlying molecular mechanisms, previous literature have suggested that Bam forms a Dub complex with Otu to control the ubiquitination and turnover of diverse target proteins (e.g., CycA and dTraf6), thus contributing to the regulation of germline development (*Ji et al., 2017*) or immune homeostasis (*Ji et al., 2019*). Between

them, dTraf6 has been shown to play a role in the maintenance of intestinal homeostaisis (*Tang et al., 2013*; *Ji et al., 2019*), making one easily reason that Bam-mediated lipid storage and energy metabolism is likely in a dTraf6-dependent manner. Unfortunately, current studies on dTraf6 are insufficient to support this notion and it would be worthwhile to test this hypothesis in future projects. Based upon our observations however, we can also not exclude that Bam may target other unknown substrate(s) to control the host metabolism. Thus, a further genetic screening using various mutant strains or transgenes (e.g., RNAi or over-expression) will be helpful to address this issue.

## CONCLUSIONS

Through examinations of the intestinal profiles and the metabolic states of *bam* loss-of-function mutant flies and *Gm114 KO* mice, we found that *bam*/*Gm114* play a conservative role in controlling the intestinal homeostasis and lipid metabolism.

## ACKNOWLEDGEMENTS

We thank the staff members at Omics-Laboratory of the Biotechnology Center for Anhui Agricultural University for providing technical supports and the other valuable discussions.

### Funding

This research was funded by Anhui Provincial Natural Science Foundation, grant number 2008085J14. The funders had no role in study design, data collection and analysis, decision to publish, or preparation of the manuscript.

### Grant Disclosures

The following grant information was disclosed by the authors:
Anhui Provincial Natural Science Foundation: 2008085J14.

### Competing Interests

The authors declare there are no competing interests.

### Author Contributions

- Jiale Wang conceived and designed the experiments, performed the experiments, prepared figures and/or tables, and approved the final draft.
- Yangyang Zhu performed the experiments, authored or reviewed drafts of the article, and approved the final draft.
- Chao Zhang performed the experiments, analyzed the data, authored or reviewed drafts of the article, and approved the final draft.
- Renjie Duan analyzed the data, authored or reviewed drafts of the article, and approved the final draft.
- Fanrui Kong analyzed the data, prepared figures and/or tables, and approved the final draft.
- Xianrui Zheng conceived and designed the experiments, authored or reviewed drafts of the article, and approved the final draft.
- Yongzhi Hua conceived and designed the experiments, performed the experiments, prepared figures and/or tables, authored or reviewed drafts of the article, and approved the final draft.

## Animal Ethics

The following information was supplied relating to ethical approvals (i.e., approving body and any reference numbers):

The Laboratory Animal Center of Anhui Agricultural University provided full approval for this research (license number: SYXK 2020-007).

## Data Availability

The raw data is available in the Supplemental Files.

## Supplemental Information

Supplemental information for this article can be found online at http://dx.doi.org/10.7717/peerj.14145#supplemental-information.

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
