# Peer review of "A conserved role of bam in maintaining metabolic homeostasis via regulating intestinal microbiota in Drosophila"

_PeerJ, doi:10.7717/peerj.14145_

## Round 0.1 · original submission · Major Revisions

I have carefully read the reviewer's comments that the article can be considered for publication after major revisions.

Reviewer 1 ·

Basic reporting

In this manuscript, authors revealed a new function of bam which involves in lipid homeostasis in the host flies.

Experimental design

Authors first noticed that bam deficient flies display increased lipid metabolites. Tissue-specific KD experiment narrowed down a potential functional contribution of bam in the gut. In the following, they found that bam deficient flies have higher bacteria load especially Bacilli and Gammaproteobacteria but not Alphaproteobacteria and this phenotype was abrogated in the axenic condition, proposing that metabolic function of bam is mediated by gut microbiota. Authors further investigated a potential contribution of Gm114, a homolog of bam, in mice. Similar to the case of flies, Gm114 KO mice have accumulated fat, and the result from DDS treated experiments indicated that Gm114 is involved in the maintenance of intestinal homeostasis.

Validity of the findings

Obviously, this is a large amount of data covering from biochemistry and genetics in Drosophila to mammalian system. Though it is definitely worth for the publication, I have still several comments. I hope that following comments will further increase the quality of this manuscript and be useful for the community.

Additional comments

Major points:
1. Fig 1A: To my knowledge, GS-MS lipidomic analysis is very powerful system detecting not only total lipids but also other detail classes of lipids such as cholesterol, ergosterol, fatty acids, fatty estels etc. Why did not authors examine these different types of lipids? Clarify the reason. If authors have more detail lipidomic profile, this data should be presented in the manuscript. Along with this line, why did authors independently measure other metabolites by classical kits?

2. Fig 5A: In this experiment, authors did not further investigate 1) if which bacteria are increased in Gm114 KO mice and 2) if antibiotic treatment recovers the phenotype. They also used SPF mice in which normally contain fewer bacteria. If additional microbiota is supplied (maybe with high calorie foods), does the phenotype get stronger? Authors need to provide additional data to support their claim more strongly. If these experiments are not feasible, provide reasonable reasons.

3. Fig 5B-E: How does the result of DDS treatment in mice corelates to the result of drosophila study and also to the metabolic function of Gm114? The result indicates that Gm114 is required for the gut homeostasis to tissue damage. No corresponding data are presented from Drosophila study. Is there any defect regarding gut epithelial cell regeneration upon pathogen infections and/or chemical reagent-induced tissue damage in bam deficient flies?


Minor points:
1. One of the weak points of this manuscript is that authors still do not address the molecular mechanism how ubiquitin-binding protein bam/Gm114 is involved in the lipid metabolism (assuming to the direction of lipolysis) and maintenance of gut microbiota. Although it may be the subject of their future study, as previous papers have already reported the involvement of bam to several signaling pathways, it will be informative if authors provide possible hypothesis in the discussion part.

2. Numbers of first and corresponding authors are different between p4 and p5. Which one is true?

3. Line 26-28: I can’t read this sentence.

4. Line 32: evolutionarily-conserved?

5. Line 106: No information about UAS-bam-RNAi flies in this section.

6. Line 121-125: Grammatical mistake in English? Please correct them.

7. Line 131: Relating to the major point 1 mentioned above, please describe more detail information how authors utilized the BinBase database, which can apparently detect variety of metabolites (https://fiehnlab.ucdavis.edu/projects/vocbinbase).

8. Line 141: Make a space between h. and Last.

9. Line 312: In addition to the expression pattern, it is better to provide a molecular information as such how Gm114 and bam is similar in terms of sequence homology and domain structure etc.

10. Line 335: in addition to the major point 3, to have a coordination between fly and mice systems, how about checking the expression of inflammation markers such as TNFα and some interleukins? Accordingly, how about checking the expression of UPDs in the gut of bam deficient flies upon tissue damage?

11. Line 377: Too many repetitions of using “Of interest”. There are five times in the manuscript.

12. Line 399: It seems that real raw data of GC-MS analysis is not provided.

Annotated reviews are not available for download in order to protect the identity of reviewers who chose to remain anonymous.

Reviewer 2 ·

Basic reporting

The study by Wang et al., is written properly, and the introduction contains sufficient to understand the context of the study. In general, the manuscript is well presented with clear and concise figures. The resolution of the figures is sufficient, the graph are well annotated, legends are concise but clear.

Experimental design

In this study by Wang et al. reports a novel function of bam in regulating metabolic homeostasis in fruit flies. The authors show that ubiquitous loss of bam or specific silence of bam in intestinal cells results in a marked increase in lipid storage and an elevated load of some types of microbiotas in the gut. A big strength of this work is the nice incorporation of two animal models (both invertebrate and vertebrate) with similar phenotypic outcomes, suggesting a conserved role of Bam/Gml14 in controlling the host metabolic state. Therefore, this study fits with the scope of the journal. The data provided by the authors are well controlled.
Authors need to provide additional information in the material and methods.
1. Line 120 – 124 : the authors should re-write this sentence, it is not clear.
2. Antibodies section: the authors should provide the reference of the antibodies
3. Line 146: The authors should mention the composition of the “cold buffer” they are using.
4. Line 174: “Intestines (from indicated flies or mice) were collected and subjected to genomic DNA extraction according to the manufacturer.s protocols.” The authors should provide the name of the kit used for the DNA extraction.
5. Line 192 - 193 : The authors explained that mice are treated with 2% of DSS during 7 days and mice are sacrificed at day-8. However, in the results section (line 332 -333), the authors wrote “we administered Gm114 KO mice and WT controls with 2% DSS in drinking water for 7 days, followed by normal drinking water for 14 days to induce a colitis model”. The authors should clarify this point.

Validity of the findings

The data provided by the authors are encouraging but further investigations need to be performed in order to have a better understanding of the mode of action bam in the gut.
1. The authors show that loss of function bam is associated with modification of microbiota, but, they don’t have more direct evidence showing how this microbiota alteration is involved in regulating host metabolic homeostasis. Can the authors provide experiments / or at least comment about the potential mechanism by which bam is associated with microbiota alterations.
2. The mouse used in this study is a whole body knock-out mice for Gm114 which is an analog of bam in the mice. Although these data show that Gm114 is essential to maintain intestinal homeostasis in mice, it is difficult to really make sure that the phenotype observed is due to the deletion Gm114 in the intestinal cells since it is a whole body knock-out mice. This information should be discuss in the discussion section. What is the expression profile of Gm114 in the mice? which cells/ organs are expressing Gm114?
3. Bam has been described as an ubiquitin associated protein important in the deubiquitination process. Can the authors provide experiments / or at least comment about the mode of action of bam in the metabolic regulation homeostasis in the gut.

Minor points:
1. Line 32, summary, remove the additional space in evolutionarily.
2. Line 126, one space is missing between the comma and 10
3. Line 271, the authors should include the detailed information of all these Gal4 strains, to show where they work.
4. Line 283, the information of bam over-expression transgenic fly is missing.
5. Lines 380, the reference of Ji et al, 2014 should be deleted.
6. Please provide statistical analyses in Figure legends not only in Materials and Methods, to help the readers better understand the whole story.

Reviewer 3 ·

Basic reporting

The authors should address the points described below.
1 The language of this manuscript should be improved. There are some typos or grammatical mistakes in this manuscript, which make it difficult to understand, the manuscript should be corrected carefully.
2 Please provide more support papers and explanations to describe how the TAG decreases during aging in Drosophila. (line 228) .

Experimental design

1 Figure 2 A and B showed that Bam expression in fat body, but this study described differently. It has detected the head, thorax, abdomen (without the gut and reproductive), gut and reproductive organ. Would you please explain which tissues you have tested?
2 Figure 4 C: Which fat tissue do you have tested? (SAT, subcutaneous adipose tissue; WAT, white adipose tissue or BAT, brown adipose tissue).
3 Please to describe more detail in Methods and Figure legend.

Validity of the findings

The presented data basically support their conclusions and would contribute to expanding the understanding of the connection between metabolic homeostasis and intestinal microbiota in Drosophila.

---

## Round 0.2 · Minor Revisions

I agree with the comments made by the reviewers, and the language should be further polished.

Reviewer 2 ·

Basic reporting

The study by Wang et al., is well presented, and the figures legends were improved as suggested with the statistical analysis. This study is suitable for publication; however, I still have some comments to improve the quality of the manuscript.
I suggest to modify the sentence in line 56-60 because it is too long and difficult to understand.
Moreover, in the materials sections, the authors indicate at line 226 “Data from Figures 4C, 4E, and 4F were obtained from 7 biological replicates. In Figures 4A and 4D, the numbers of experimental mice were 38 for WT and 35 for Gm114 KO.” This information is different in the legends where the authors wrote n=3. Which information is the right one? Can you clarify?

Experimental design

The authors modified the materials and methods as suggested. However, some points need to be clarified.
- Line 145: It is surprising that PBS is sufficient to release proteins, lipids and others metabolites.
- Line 222: in the methods it is written “LogRank test in the PASW Statistics 18 software in Figure 5A”. However, in the figure legends, this statistical analysis was used for Figure 5B.

Validity of the findings

The authors modified the discussion especially with the last sentence for the mouse model. I understand that it is complicated and long to obtained this specific mouse model.
Regarding my comment N°3 “Bam has been described as an ubiquitin associated protein important in the deubiquitination process. Can the authors provide experiments / or at least comment about the mode of action of bam in the metabolic regulation homeostasis in the gut.” Unless I am mistaken, the authors did not really discuss about it. I was expecting to have some hypotheses in the discussion. For example, in the introduction it is written “In-depth studies further unraveled the biochemical natures of Bam protein as a ubiquitin associator (Ji et al., 2017) and positively regulates the deubiquitination processes of specific ubiquitinated targets (Ji et al., 2017; Ji et al., 2019).” (introduction, line 78 – 80). What is known about these specific ubiquitinated targets? Are they important in the gut homeostasis? Maybe by checking in the literature, the authors can provide some ideas in the discussion.

However, I think this study is suitable for publication as the results described in this manuscript will be useful for the scientific community.

Minor points:
1. Line 245: “adults were apparently higher”, should be “significantly” higher
2. Line 319 : “more than 1 fold”, according to the figures, it is more than 2 fold.
3. Line 339 : “when we raised..” the capital letter for When is missing
4. Figure 5A : I agree that there is an upward trend of the gut microbiota, but I am surprised that the results are significant regarding this such big standard deviation.
5. Legend figure 5 : scale bars 40mm ?

Reviewer 3 ·

Basic reporting

There are still some typos or grammatical mistakes in this manuscript, such as line 19.

Experimental design

no comment

Validity of the findings

I have a suggestion if it is better to talk about the previous study published in
Front Cell Dev Biol in 2022 in your Figure 5 or Discussion.

---

## Round 0.3 · accepted · Accept

The authors have made revisions in accordance with their comments, and the current manuscript version is recommended for publication in PeerJ.